# Multivariate resilience indicators to anticipate vector-borne disease outbreaks: A West Nile virus case-study

Clara Delecroix[1,2]*, Quirine ten Bosch[2], Egbert H. Van Nes[1], Ingrid A. van de Leemput[1]

**1** Environmental Sciences Group, Wageningen University & Research, Wageningen, The Netherlands,
**2** Infectious Disease Epidemiology, Wageningen University & Research, Wageningen, The Netherlands

\* clara.delecroix@wur.nl

## Abstract

### Background and aim

To prevent the spread of infectious diseases, successful interventions require early detection. The timing of implementation of preventive measures is crucial, but as outbreaks are hard to anticipate, control efforts often start too late. This applies to mosquito-borne diseases, for which the multifaceted nature of transmission complicates surveillance. Resilience indicators have been studied as a generic, model-free early warning method. However, the large data requirements limit their use in practice. In the present study, we compare the performance of multivariate indicators of resilience, combining the information contained in multiple data sources, to the performance of univariate ones focusing on one single time series. Additionally, by comparing various monitoring scenarios, we aim to find which data sources are the most informative as early warnings.

### Methods and results

West Nile virus was used as a case study due to its complex transmission cycle with different hosts and vectors interacting. A synthetic dataset was generated using a compartmental model under different monitoring scenarios, including data-poor scenarios. Multivariate indicators of resilience relied on different data reduction techniques such as principal component analysis (PCA) and Max Autocorrelation Factor analysis (MAF). Multivariate indicators outperformed univariate ones, especially in data-poor scenarios such as reduced resolution or observation probabilities. This finding held across the different monitoring scenarios investigated. In the explored system, species that were more involved in the transmission cycle or preferred by the mosquitoes were not more informative for early warnings.

**Data availability statement:** There are no primary data in the paper; all materials are available at https://doi.org/10.5281/zenodo.15680364.

**Funding:** This publication is part of the project 'Preparing for Vector-Borne Virus Outbreaks in a Changing World: a One Health Approach' (NWA.1160.18.210), which is (partly) financed by the Dutch Research Council (NWO) (all authors). The funders had no role in study design, data collection and analysis, decision to publish, or preparation of the manuscript.

**Competing interests:** The authors have declared that no competing interests exist.

## Implications

Overall, these results indicate that combining multiple data sources into multivariate indicators can help overcome the challenges of data requirements for resilience indicators. The final decision should be based on whether the additional effort is worth the gain in prediction performance. Future studies should confirm these findings in real-world data and estimate the sensitivity, specificity, and lead time of multivariate resilience indicators.

## Author summary

Vector-borne diseases (VBD) represent a significant proportion of infectious diseases and are expanding their range every year because of among other things climate change and increasing urbanization. Successful interventions against the spread of VBD require anticipation. Resilience indicators are a generic, model-free approach to anticipate critical transitions including disease outbreaks, however the large data requirements limit their use in practice. The transmission of VBD involves several species interacting with one another, which can be monitored through different data sources. The information contained by these different data sources can be combined to calculate multivariate indicators of resilience, allowing a reduction of the data requirements compared to univariate indicators relying solely on one data source. We found that such multivariate indicators outperformed univariate indicators in data-poor contexts. Multivariate indicators could be used to anticipate not only VBD outbreaks but also other transitions in complex systems such as ecosystems' collapse or episodes of chronic diseases. Adapting the surveillance programs to collect the relevant data for multivariate indicators of resilience entails new challenges related to costs, logistic ramifications and coordination of different institutions involved in surveillance.

## Introduction

To prevent the spread of infectious diseases, successful interventions require early detection. The timing of implementation of preventive measures is crucial, but as outbreaks are hard to anticipate, control efforts often start too late. The use of resilience indicators to anticipate infectious disease outbreaks has attracted attention in the past decade: they are generic, model-free indicators used to anticipate critical transitions in complex systems [1,2]. These indicators typically rely on the theory of critical slowing down, stating that a system approaching a critical transition, such as the start of an epidemic, loses its resilience and recovers more slowly from external perturbations [3–6]. In the context of infectious diseases, this translates into a longer time for minor outbreaks to resolve. This slow behaviour can be observed in trends over time of

so-called resilience indicators, namely autocorrelation and variance [7]. Such trends are usually evaluated in the prevalence time series using a rolling window, and their significance can be assessed by calculating Ebisuzaki p-values [8]. Resilience indicators have proven to be able to anticipate upcoming epidemics up to several months in advance for infectious diseases, including mosquito-borne diseases [1,9–11]. However, the large data requirements limit their use in practice, it is thus necessary to design appropriate monitoring strategies consistent with the disease system of interest.

Surveillance of multi-host diseases is challenging to achieve, as many, sometimes unknown, species can participate in the transmission, which leaves choices on which sources to sample from [12]. For instance, West Nile virus, an enzoonotic disease transmitted between birds and mosquitoes but also affecting horses and humans as dead-end hosts, can be monitored in numerous ways. Transmission can be monitored in mosquito pools by estimating the proportion of infected mosquitoes or the proportion of infected pools [13], but also by monitoring infections in humans, livestock, or wildlife through public health and veterinary surveillance systems [14]. Choices then must be made on where to direct sampling efforts. Collecting data from multiple sources would either increase the sampling costs and efforts, or reduce the quality or sampling frequency of each data source. Contrarily, focusing on one single data source enables the relevant authorities to concentrate all their efforts on sampling one source meticulously, but the information contained in other sources may be missed. Additionally, when several species act as enzootic hosts and react differently to the disease, one can wonder which species is the most informative for early warnings of future outbreaks.

When data from multiple sources are available, the information they contain can be combined to calculate multivariate indicators of resilience, for instance mean autocorrelation and mean variance [15]. Previous research showed that multivariate indicators signal upcoming critical transitions in the same way univariate resilience indicators do, using simulated plant-pollinator data with a generic model [15]. These multivariate indicators rely on data reduction techniques to combine multiple data streams, such as Principal Component Analysis (PCA) and Maximum Autocorrelation Factors (MAF). Common resilience indicators such as autocorrelation and variance are then computed in the obtained combined time series. Similar multivariate indicators have been investigated in simulated prevalence time series of metapopulation models and could signal upcoming epidemics [16]. However, they have not been compared to univariate indicators. As multivariate indicators require a greater monitoring effort, researching the best monitoring strategy would be of use to inform monitoring practices.

That question is not limited to infectious disease outbreaks: anticipating critical transitions in multivariate systems is relevant for a diversity of research areas, such as ecology [17,18] or finance [19]. Designing monitoring strategies for complex systems is complicated by the numerous entities interacting, sometimes with poorly understood interactions [20]. Additionally, some entities seemingly affecting the dynamics of the system more directly are not always the easiest to monitor. For instance, food webs or ecosystems that are nearing collapse often contain many interacting species, for which central endangered species are not always easy to monitor [20]. Previous research showed that some species can display more reliable [21] and earlier [22] signals using resilience indicators as early warning, but this has not been studied in epidemiology.

West Nile virus (WNV) is a relevant example of a multi-host, vector-borne disease (VBD). While human and horse cases can seem easy to monitor via public health and veterinary authorities, the small proportion of symptomatic cases results in low reporting probabilities [14,23]. On the other hand, monitoring wild birds is often done by collecting and analyzing dead birds [24]. Monitoring live birds is also possible but requires a greater effort as they must be caught and sampled by bird ringers or citizen scientists, e.g., [25–27], and is thus more costly than sampling dead birds. Alternatively, hunting or trapping also allows for live bird sampling [28]. Furthermore, the species contributing to the transmission of VBDs like WNV are not all known. Sometimes an unknown, hidden (i.e., not dying from the disease) reservoir can be central to the transmission dynamics [29,30]. Infected mosquito pools can be monitored via traps, but estimating the prevalence of WNV in mosquitoes is complicated by logistic and cost limitations [31]. Finally, as the transmission is dependent on the ecosystem setting, such as the abundance of the different hosts and vectors as well as vector feeding preferences, changes in ecosystems can impact the transmission if, e.g., several host species or vector species compete [32], making the task of identifying which species contribute most to transmission complex and context-specific. It is then relevant to know how to identify informative data sources to direct sampling efforts to anticipate future WNV outbreaks.

In the present study, we compare multivariate and univariate indicators of resilience under different monitoring scenarios as an early warning for upcoming epidemics of vector-borne diseases. We use WNV as a case study, as it affects multiple host and vector species, and can be monitored in numerous ways. A synthetic dataset is generated using a simple, well-studied WNV model. We explore several monitoring scenarios focusing on different data types, including data-poor scenarios with low reporting probability or resolution, and try to identify what makes a species more or less informative for early warnings.

## Materials and methods

### Model

We use a well-known compartmental Ross Macdonald model of WNV for which the dynamics are well-studied [33]. Our model considers the vector population and four host populations: two representative bird species acting as amplifying hosts, and humans and horses acting as dead-end hosts (Fig 1A, equations provided in S1 Text). The two bird species differ in how they are affected by the disease. Bird species V is a visible, i.e., detectable, reservoir for the disease: 30% of the infected individuals die from infection, leading to high number of dead birds but low seroprevalence (as can be the case for corvids, for instance [30,34]). Bird species H is a hidden reservoir: it is preferred by mosquitoes but it does not experience symptoms and does not die from infection, hence complicating the detection of transmission in this species through dead bird surveillance. However, both bird species are equally hard to monitor through live bird surveillance. The model is used to generate time series.

The mosquito population is subdivided into susceptible mosquitoes $M_S$, exposed mosquitoes $M_E$, and infectious mosquitoes $M_I$. Mosquitoes get infected with a probability $p_M$ when biting infectious birds, which they encounter with a rate $k$, denoting the biting rate of mosquitoes. Additionally, we introduce biting preference coefficients $p_i$ for each host species: bird H is the preferred host, followed by bird V, and humans and horses. We assume that infected mosquitoes do not recover from the infection and that they are not affected by an additional disease-induced death rate. The mosquito population is in equilibrium as the birth rate and the death rate $b_M$ are considered equal.

The bird, human, and horse populations are subdivided into susceptible (respectively $B_{VS}$, $B_{HS}$, $H_S$ and $E_S$), exposed (respectively $B_{VE}$, $B_{HE}$, $H_E$ and $E_E$), infectious (respectively $B_{VI}$, $B_{HI}$, $H_I$ and $E_I$), and recovered (respectively $B_{VR}$, $B_{HR}$, $H_R$ and $E_R$). The bird and horse populations are assumed to be at equilibrium in the absence of disease as the natural birth rate and the natural death rate (respectively $b_B = m_B$ and $b_E = m_E$) are considered equal. The natural death rate affects all compartments of birds and horses equally. They are affected by an additional disease-induced death rate $v_{VA}$, $v_{HB}$, and $v_E$, respectively, and the count of dead birds V over time is reported in a compartment $B_{VD}$, which is the cumulative number of dead birds V and thus only increases over time. Additionally, WNV is maintained in the population via stochastic events of importation of infectious birds at a rate $\Lambda_B$. We consider that the human population dynamics are insignificantly slow compared to the disease dynamics, and we do not include them in the equations. The transmission rate by infectious mosquitoes is respectively $p_M k p_{BV}$, $p_M k p_{BH}$, $p_M k p_H$, and $p_M k p_E$ (with $p_M$ the mosquito-to-host transmission probability, $k$ the biting rate of mosquitoes, and $p_{BV}$, $p_{BH}$, $p_H$ and $p_E$ mosquito biting preference coefficients).

We use the parameter values provided in [33] and [35] as default values (section Model details in S1 Text). Intrinsic stochasticity arising from random variation in the timing of events is added to the model using the Gillespie algorithm [36]. Simulations are run in R 4.2.3 using the package SimInf [37].

### Perturbation-recovery experiments

To study the presence of critical slowing down in this system, we perform perturbation-recovery experiments using the WNV model described above for different values of $R_0$ (i.e., the basic reproduction number, defined as the number of secondary cases expected from an infectious individual in a fully susceptible population). $R_0$ is calculated using the Next Generation Matrix (NGM) (section Model details in S1 Text) [38], and the biting rate values are chosen to obtain the desired $R_0$ values in the simulations described below. Five infectious individuals are introduced in the disease-free population at $t = 5$,

 

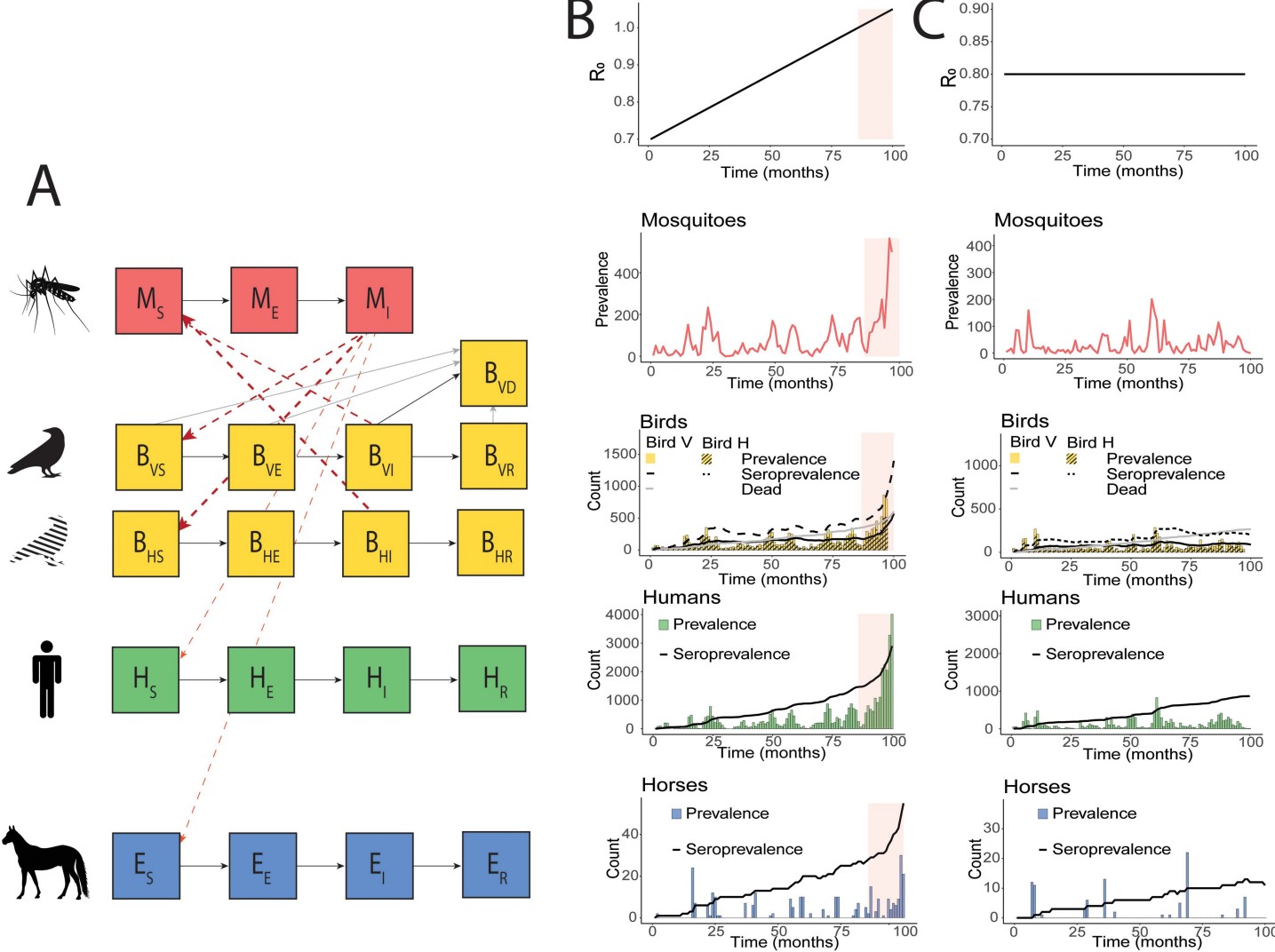

**Fig 1. WNV model and simulated data.** (A) WNV compartmental model used in the analyses. (B) Example of a simulated time series with an upcoming epidemic, simulated by increasing $R_0$ from 0.7 to 1. The shaded region indicates where $R_0$ is above 1. (C) Example of a simulated time series with no upcoming epidemic, simulated with a stable $R_0 = 0.8$. In (B) and (C), the time series represent a single simulation and were aggregated per month to improve the readability of the figure, but the daily time series are used in the analyses.

and the recovery time to the disease-free equilibrium (i.e., the absence of infectious birds, resp mosquitoes) is calculated based on the exponential decay of infectious individuals after the perturbation. The slope of a linear regression between the log number of infectious individuals and time with a fixed intercept is used to calculate the decay rate. The reciprocal of this number is the recovery time from a perturbation. We perform these experiments for both birds and mosquitoes, over 1000 replicates per $R_0$ value to estimate the average and the confidence intervals of the recovery time.

## Data simulations

For our analyses, we assume that mosquito prevalence, bird prevalence and seroprevalence, human prevalence and seroprevalence, and horse prevalence and seroprevalence could be monitored, initially with high precision.

The biting rate varies over time (staying in the plausible interval 0.2-0.92 suggested in [35]), altering the $R_0$ and thus the resilience of the system. The biting rate is known to be influenced by temperature, which mimics a simplified representation of the emergence of WNV due to increasingly suitable climatic conditions over the years [39]. Other parameters affected by the temperature, such as the extrinsic incubation period or mosquito birth and death rates, are kept fixed to avoid interactions between the parameters obscuring the dynamics of emergence, and thus our understanding of the indicators. For the same reasons, seasonal fluctuations were omitted and the emergence of WNV was modelled as being driven solely by the between-year increase of the average $R_0$ associated with climate change.

We simulate an approaching epidemic due to increasingly suitable conditions in single long runs, with the biting rate increasing slowly over time. These so-called "emergence time series" are used to measure the performance of the indicators as an early warning for upcoming epidemics. In these time series, $R_0$ gradually increased from 0.7 to 1 over 428 weeks with a weekly resolution (Fig 1B). As a control, we simulate "stable time series" under the assumption of a fixed $R_0$ over time, leading to no major outbreaks (Fig 1C). We use $R_0 = 0.8$ for that scenario. We simulate 100 stochastic repetitions for each $R_0$ condition, and the full simulated time series were used in the analyses.

## Monitoring scenarios

We consider 10 data sources to be used in our early-warning signals: mosquito, bird species V, bird species H, human and horse prevalence time series represented by model states $M_I$, $B_{VI}$, $B_{HI}$, $H_I$ and $E_I$, bird, human and horse seroprevalence time series represented by the states $B_{VR}$, $B_{HR}$, $H_R$ and $E_R$ as well as time series of dead birds species V represented by the state $B_{VD}$. We considered these time series individually to evaluate the performance of univariate resilience indicators. For multivariate indicators, three scenarios representing different data combinations are created. Scenario A represents an anthro-equine scenario, where all human and horse data (prevalence and seroprevalence) are being collected ($H_I$, $E_I$, $H_R$, $E_R$). Scenario B represents a wildlife scenario where all mosquito and visible (i.e., detectable) bird data (prevalence, seroprevalence and deaths) are being collected ($M_I$, $B_{VI}$, $B_{VR}$, $B_{VD}$). Finally, scenario C represents a hidden bird species investigation where the hidden bird species data is collected along with mosquito data ($M_I$, $B_{HI}$, $B_{HR}$).

To try to identify what affects the prediction performance of a given species, we vary the feeding preference coefficient, affecting how much each species gets infected by a typical infectious individual. We generate time series of the model with six values for the feeding preference coefficient of mosquitoes towards bird species H. The feeding preference coefficient towards bird H is chosen as it has a direct, large effect on how much all the species get bitten by mosquitoes, and thus can get infected (Fig L in S1 Text). Additionally, it allows to explore how potential hidden reservoirs could obscure early-warning signals. For each feeding preference scenario, we quantify how much each species gets infected by a typical infectious individual, called the "*typical infection coefficient*" in the rest of this manuscript (Fig L in S1 Text). To estimate the *typical infection coefficient*, we use the corresponding coefficient in the eigenvector associated with the dominant eigenvalue of the NGM (Section model details in S1 Text). In the default emergence time series, bird H is preferred by the mosquitoes with a coefficient $p_{BH} = 10$, against $p_{BV} = 5$ for bird V, and $p_E = p_H = 1$ for horses and humans. Additionally, we generate time series with $p_{BH} = 5$, 20, 30, 50, and 80. For each simulation, we adapt the range of values for the biting rate to keep a range of $R_0$ going from 0.7 to 1 for the emergence time series and $R_0 = 0.8$ for the stable time series. The feeding preference coefficient affected the typical infection coefficient of all species differently, for instance it made little difference for horses as they are not densely highly abundant (Fig L in S1 Text).

To investigate the limitations of resilience indicators as an early warning sign for upcoming epidemics in data-poor scenarios, we explore the effect of two additional complexities: (a) reduced resolution, and (b) reduced observation probability. For (a), the data are down-sampled from the full emergence time series by reducing the resolution. Thus, the $R_0$ gradient remains unchanged, but the number of data points decreases. Secondly, (b) was explored using the full emergence

time series. For each data point, observation noise was added using a Poisson distribution of expectation $\lambda = number\ of\ observations$ for that time step for that variable. To reproduce imperfect observations, we define three observation probabilities: $p_1 = 0.1$, $p_2 = 0.01$, and $p_3 = 0.001$. Imperfect observations are then generated for each data point using a Poisson distribution of expectation $\lambda = p_i\ x\ number\ of\ observations$.

## Resilience indicators

We calculated all resilience indicators using the rolling window method for time series as described in [7]. For this method, we need to detrend the data to avoid spurious increase of the indicators due to their trend. We used a Gaussian kernel with the optimal bandwidth according to [40] for this (section Detrending of the time series in S1 Text). A rolling window of 50% of the size of the detrended time series is used to calculate the indicators and assess their trend over time, reflecting the most common choice for the size of the rolling window [2]. All complete windows are used, with the first spanning from the start to the midpoint of the dataset, and the last spanning from the midpoint to the end. In each window, the value of each indicator is calculated (for univariate indicators, using the built-in functions to calculate variance and autocorrelation in Matlab) [7]. We then evaluate the strength of the trend in the indicators over time by calculating the Kendall tau correlation between the value of each indicator in each window and time. We use the open-source tool generic_ews for Matlab (https://git.wur.nl/sparcs/generic_ews-for-matlab) [41]

We use a subset of the multivariate indicators described in [15]: mean autocorrelation and variance, maximum autocorrelation, covariance, cross-correlation and variance, PCA and MAF autocorrelation and variance, and PCA eigenvalue and MAF eigenvalue. These indicators rely on different techniques to combine multiple time series. Some indicators use the maximum or average of a univariate indicator among all time series. For instance, the maximum variance (Max Var) and maximum autocorrelation (Max AR) calculate the variance (resp. autocorrelation) for all variables but keep only the maximum value of variance (resp. autocorrelation) for each window. Similarly, the mean variance (mean var) and mean autocorrelation (mean AR) take the mean of variance (resp. autocorrelation) over all variables for each window. The maximum covariance (max cov) is the maximum coefficient of the covariance matrix, and the mean cross-correlation (max abscorr) is the mean of all the absolute correlations between the variables. Furthermore, two more elaborate data reduction techniques were used to combine all the time series (PCA and MAF). PCA indicators are calculated in time series of the system projected in the direction of highest variance, corresponding to the first component of a principal component analysis. The maximum eigenvalue of the covariance matrix used in the PCA is equal to the amount of variance explained by the first principal component (expl var). The variance (PCA var) and autocorrelation (degenerate fingerprinting, Deg Finger) are then calculated in the projected time series. Similarly, the direction of highest autocorrelation is identified using an MAF analysis (Max Autocorrelation Factor analysis, section Multivariate Factor Analysis in S1 Text) [42]. The minimum eigenvalue of the MAF analysis (MAF eig) is calculated for each window, and variance (MAF var) and autocorrelation (MAF AR) were calculated for the projected time series. We compare the performance of these multivariate indicators with the most common univariate indicators, namely autocorrelation and variance of each single time series.

We quantify the indicators' performance as early-warning signals using ROC (receiver operating characteristic) curves. Emergence time series with increasing $R_0$ leading to an epidemic are used to calculate the true positive rate and false negative rate, while time series with a fixed $R_0$ with no upcoming epidemic are used to calculate the true negative and false positive. The ROC curve is calculated based on each value of Kendall tau from the 100 simulations with an upcoming epidemic and 100 simulations with no upcoming epidemic. Each value of Kendall tau is labeled with the outcome (upcoming epidemic or not), and the ROC curve is implemented using the perfcurve function of the Statistics and Machine Learning toolbox in Matlab. The Area Under the ROC Curve (AUC) was used to estimate the prediction performance of the different indicators under different data scenarios over 200 repetitions, 100 emergence time series and 100 fixed time series.

## Results

### Perturbation recovery experiments

To investigate the presence of critical slowing down in the model, perturbation recovery experiments (Fig 2A) were performed by introducing infectious birds (Fig 2A and 2B) and mosquitoes (Fig 2C) to disturb the disease-free equilibrium. In both cases, the recovery time increased as $R_0$ approached the critical threshold, indicating a loss of resilience as the system approaches suitable conditions for an epidemic to spark.

### Performance of the indicators as an early warning for upcoming epidemics

The Area Under the ROC Curve was used as a measure of the performance of the different indicators and scenarios, as it considers both the true positive rate (ability to correctly predict an upcoming epidemic) and the true negative rate (ability to predict that no upcoming epidemic is approaching). For univariate time series, we measured the performance for each time series and univariate indicator (Fig 3A). For multivariate time series, we measured the prediction performance for each monitoring scenario and multivariate indicator (Fig 3B–3D). The performance of univariate indicators ranges between 0.68 (Horse seroprevalence with autocorrelation) and 0.88 (Human prevalence with variance). The performance of multivariate indicators ranges between 0.56 (Anthro-equine scenariomean, ai with explained variance) and 0.94 (Hidden reservoir scenario with mean variance). Although the Hidden reservoir scenario yielded the best prediction performance across all indicators, there was a small difference in the average AUCs of the different multivariate scenarios, ranging between 0.80 and 0.83. Interestingly, the difference in average prediction performance between the univariate indicators and the multivariate ones was only 0.02 (AUC of 0.79 for univariate indicators and 0.81 for multivariate, average across all scenarios and indicators), especially considering that multivariate indicators use up to four times more data points than the univariate ones, which requires a greater monitoring effort.

Variance and variance-based indicators outperformed autocorrelation and autocorrelation-based indicators for both univariate and multivariate monitoring scenarios. For univariate time series, variance yielded an average AUC of 0.86 over all the time series, against an average AUC of 0.73 for autocorrelation. Similarly, for multivariate indicators, variance-based indicators yielded an

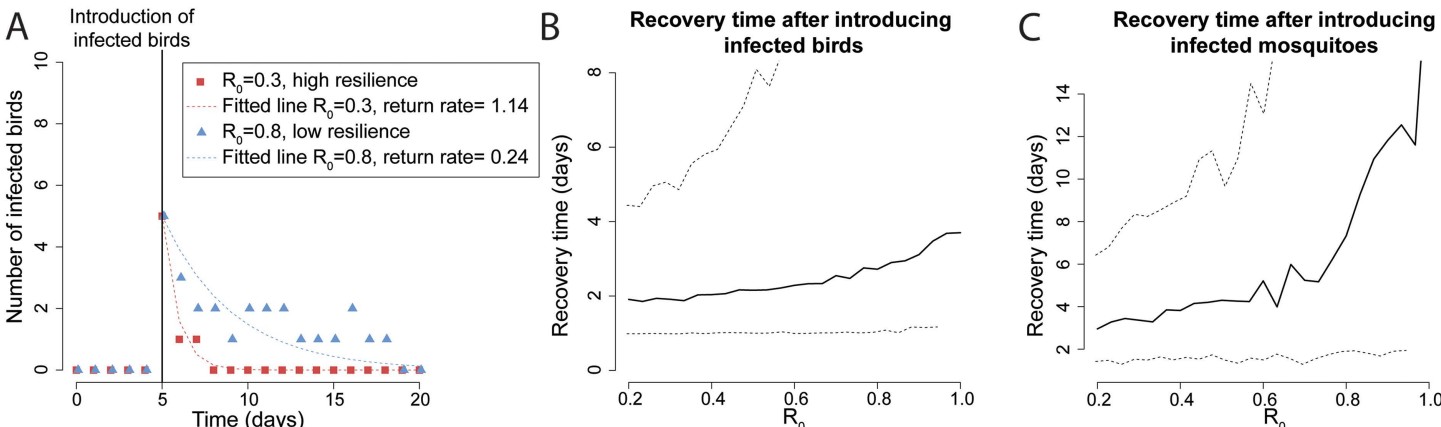

**Fig 2.** (A) Number of infected birds A over time, for two stochastic simulations as examples of the perturbation-recovery experiments in a case of high (red) and low (blue) resilience. The points represent the observations of infected birds over time, and the dotted lines indicate the fitted lines used to calculate the return rate to the disease-free state (see Methods). (B) Average recovery time (solid line) and 95% confidence interval (dotted line) (in days) to the disease-free state after perturbing the system by introducing infected birds for different values of $R_0$. (C) Average recovery time (solid line) and 95% confidence interval (dotted line) (in days) to the disease-free state after perturbing the system by introducing infected mosquitoes for different values of $R_0$. For B and C, the recovery time is defined as the reciprocal of the return rate, assuming exponential decay, and is calculated over 1000 replicates.

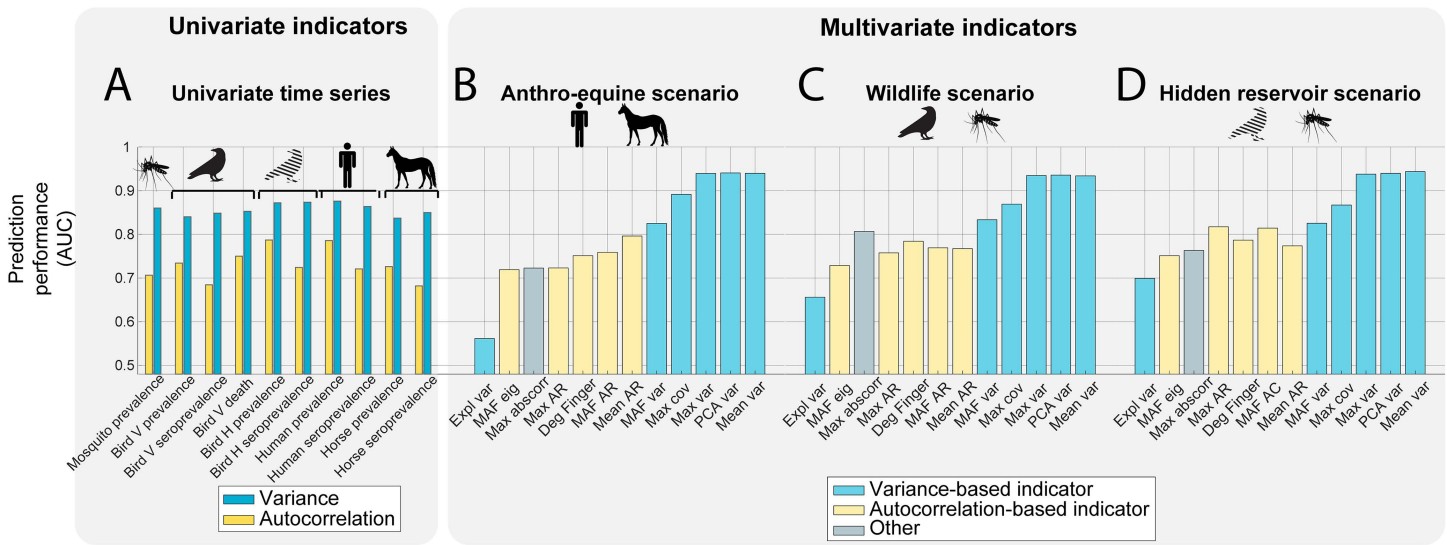

**Fig 3. Prediction performance of the different univariate and multivariate indicators of resilience evaluated using the AUC.** (A) Prediction performance of the univariate indicators for all univariate time series. (B) Prediction performance of the multivariate indicators for the Anthro-equine scenario. (C) Prediction performance of the multivariate indicators for the wildlife scenario. (D) Prediction performance of the multivariate indicators for the hidden reservoir scenario.

average AUC of 0.86 against an average AUC of 0.76 for autocorrelation-based indicators, and the three best-performing multivariate indicators were variance-based (mean variance, PCA variance and max variance). Due to the scarcity of cases far from the epidemic threshold, the time series of all hosts and vectors contained long stretches of consecutive zeros, resulting in a highly variable autocorrelation, which altered the performance of autocorrelation and autocorrelation-based indicators (Fig A to C in S1 Text).

For the rest of the analyses, we proceeded with the three best-performing indicators (variance for univariate indicators, and mean variance, PCA variance and maximum variance for multivariate indicators), the best multivariate scenario (hidden reservoir), and the best univariate time series (mosquito prevalence, bird H prevalence, seroprevalence, and human prevalence). The results of the other indicators and scenarios are provided in the S1 Text.

## Data-poor scenarios

The performance of univariate indicators of resilience was compared to the performance of multivariate indicators of resilience in down-sampled time series (Fig 4A) and for lower reporting probabilities (Fig 4B). The comparison was performed for different numbers of data points obtained by subsampling the original time series by reducing the resolution (Fig 4A and 4C, other scenarios and indicators Fig D to G in S1 Text) and for different observation probabilities implemented using a Poisson distribution (Fig 4B and 4D, other scenarios and indicators Fig H to K in S1 Text). The prediction performance of all indicators and monitoring scenarios decreased for lower numbers of data points and lower observation probabilities. Multivariate indicators demonstrated greater robustness than univariate indicators, maintaining higher performance despite reductions in data resolution and observation probability. Autocorrelation-based indicators were more sensitive to a reduced resolution, consistent with previous findings [7] (S1 Text).

## Variability in the prediction performance of the different species

We explored the nature of the variations in the prediction performance of the different monitoring scenarios observed in the previous section. Specifically, we explored how informative the different species are for early warnings depending

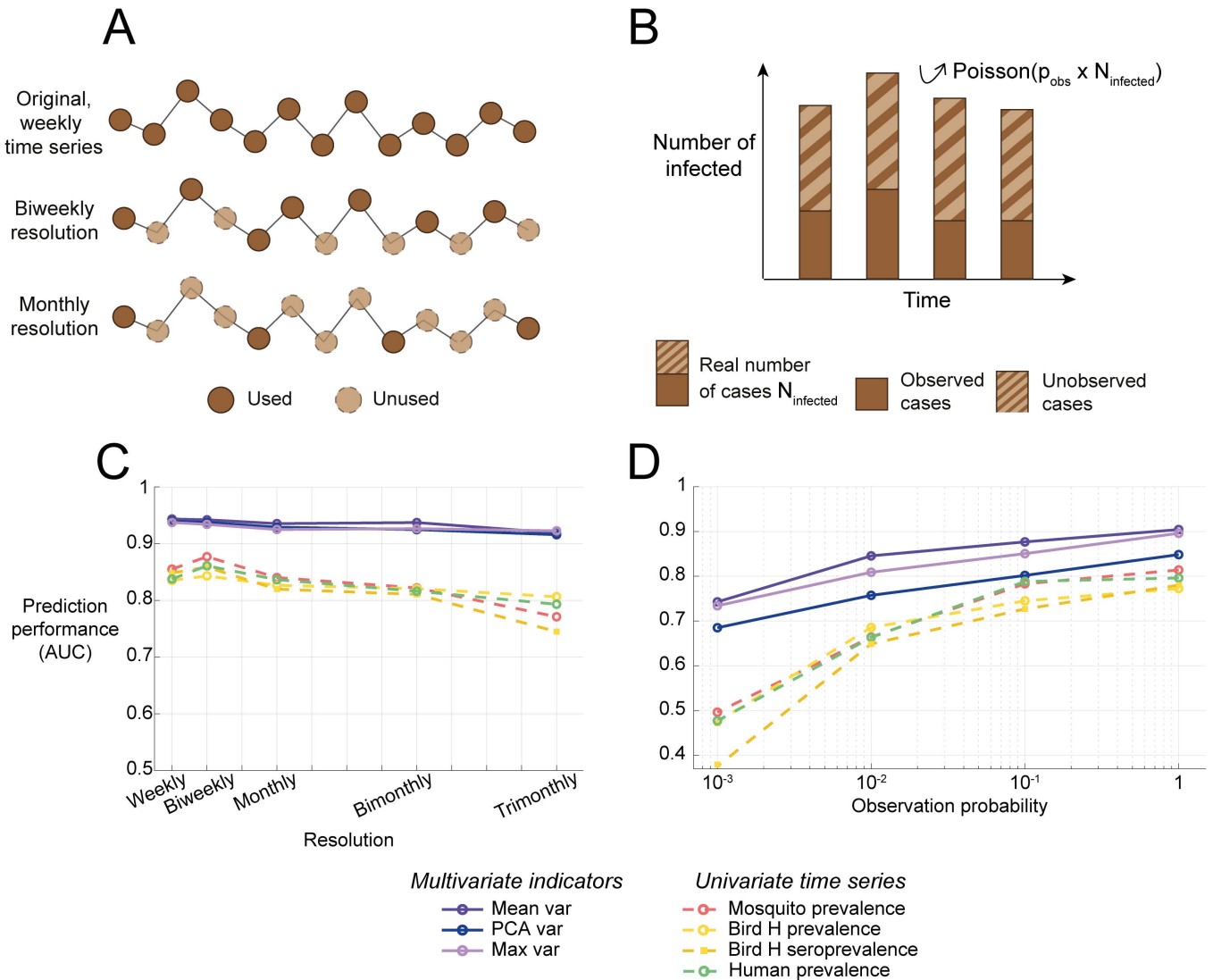

**Fig 4. Performance of the best-performing, variance-based indicators of resilience in data-poor scenarios.** (A) The resolution is reduced by downsampling the original time series. (B) The observation probability is reduced by subsampling the number of infected using a Poisson distribution. (C) Effect of the reduction of resolution on the prediction performance. (D) Effect of the reduction of observation probability on the prediction performance.

on how much they get infected in the transmission cycle. A variation in the mosquito feeding preference towards bird H impacts the *typical infection coefficient* of all the species, i.e., how much they get infected, quantified using the corresponding value in the eigenvector associated with the dominant eigenvalue of the NGM (Fig L in S1 Text). When the feeding preference towards bird H increased, the prediction performance slightly decreased for univariate indicators uniformly for all species. However, it remained stable for variance-based multivariate indicators (Fig 5A, other scenarios and indicators Fig M to P in S1 Text). Additionally, the prediction performance of the different univariate time series was unrelated to the *typical infection coefficient* (Fig 5B, other scenarios and indicators Fig Q in S1 Text). These results were robust to a reduced observation probability and resolution (Fig R and S in S1 Text). Additionally, to make sure that the choice of the varying parameter does not impact the result, the same analyses were repeated by varying the relative

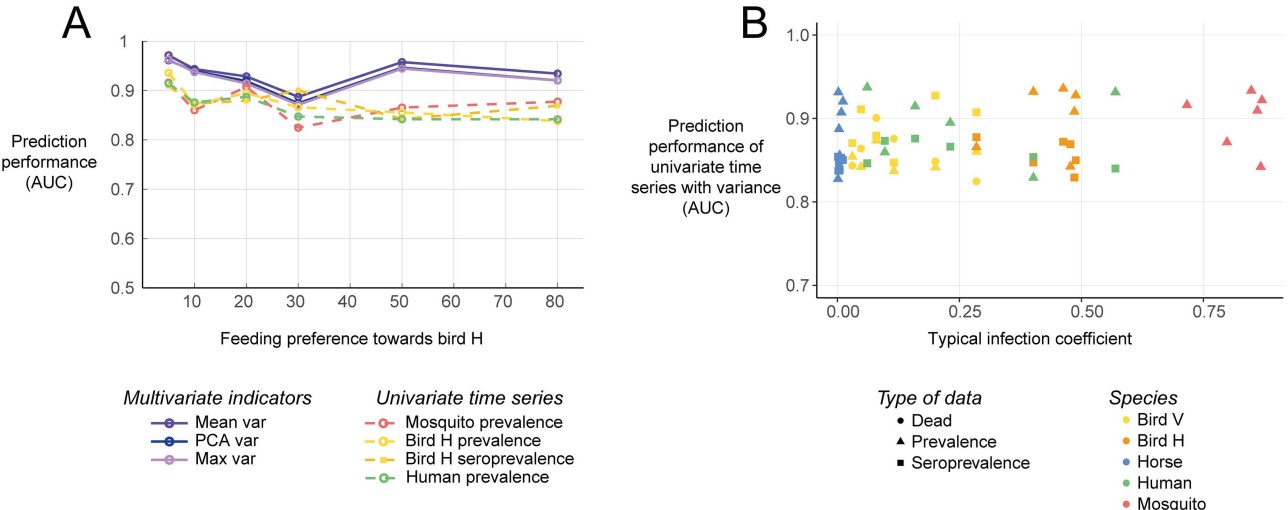

**Fig 5. Prediction performance of the different species depending on how much they get infected.** (A) Effect of varying mosquito feeding preferences towards the hidden reservoir on the prediction performance. (B) Prediction performance of the univariate time series depending on how much the species get infected, quantified using the *typical infection coefficient* (i.e., the corresponding coefficient of the eigen vector associated with the dominant eigenvalue of the NGM).

abundances of bird H to bird V instead of the feeding preference. These analyses yielded analogous results (Fig T to Y in S1 Text).

## Discussion

Using a synthetic dataset, we found that both univariate and multivariate resilience indicators could signal upcoming outbreaks of WNV. Multivariate indicators outperformed univariate indicators, especially in data-poor scenarios. The accuracy of multivariate indicators ranged up to 0.94, and those of univariate indicators up to 0.88, assuming perfect observation (Fig 3). When reducing the resolution or the observation probability, the difference in prediction performance between univariate and multivariate indicators broadened, suggesting that multivariate indicators are more robust to a decrease in the data quality (Fig 4). However, multivariate indicators require up to four times more data points as several hosts have to be monitored simultaneously. As data requirements limit the use of resilience indicators in practice, these findings could have important implications for the surveillance of vector-borne diseases. As the model was kept purposefully generic, these results can be extended to other vector-borne diseases. However, future research should confirm the findings in data, including more complexities such as seasonality [39] or complex observation processes [40], as well as cost estimations and logistic ramifications of the different monitoring strategies [24].

The prediction performance of the different multivariate monitoring scenarios, as well as univariate time series, remained stable regardless of how much a species is involved in the transmission cycle. Noticeably, increasing the mosquito feeding preference towards a given species or the relative abundances of the different species did not increase the potential for that species to signal upcoming outbreaks (Fig 5A). Thus, the variability in the prediction performance of the different monitoring scenarios could not be explained by the typical infection coefficient, namely how much a species gets infected by a typical infectious individual (Fig 5B), nor by additional properties of the time series (Fig AD in S1 Text). In contrast to earlier findings, in this system, we found no evidence that species that are central to the transmission cycle are more informative for early warning [18,22]. The model assumed relatively high spillover rates to dead-end species leading to cases in dead-end species that are observed even with low reporting probabilities. However, this is not always the case

in practice and could hamper the prediction performance of resilience indicators when no cases are detected [41]. Additional research is needed to better understand what drives how informative a data source is in anticipating future disease outbreaks using resilience indicators. Answering this question is especially relevant for the surveillance of various infectious diseases, especially when silent transmission is ongoing in hidden reservoirs that are harder to observe as they do not experience symptoms and deaths, and have to be monitored through live bird sampling [30,43].

Variance-based indicators outperformed autocorrelation-based indicators and remained robust in data-poor contexts. This is consistent with previous studies that identified variance or related indicators as the best-performing indicators when researching the performance of resilience indicators to anticipate epidemics [9,44,45]. In the present study, the poor performance of autocorrelation-based indicators was likely due to the excess of zeros, mainly when $R_0$ is far from the critical threshold, leading to a high autocorrelation and thus obscuring the overall trend in autocorrelation (Fig A to C in S1 Text). Additionally, the performance of autocorrelation-based indicators decreased when the resolution of the time series was reduced, in agreement with previous studies [15,46] (Fig D to F in S1 Text). Further, we kept regular sampling intervals in our analyses, as this is necessary for the calculation of autocorrelation. Irregular sampling interval would require interpolation which would probably further reduce the performance of autocorrelation based-indicators. Multivariate variance-based indicators of resilience showed stable performance, even when reducing the resolution of time series or the observation probability (Fig 4). This was especially true for indicators using the average of univariate indicators, consistent with previous research [15]. Contrastingly, explained variance performed particularly poorly, probably because of the reactivity of the system leading to fluctuations in a different direction than the direction of maximum variance, evaluated with the PCA [18]. As data requirements are a major factor limiting the use of resilience indicators in practice, multivariate indicators hold potential to improve the performance of resilience indicators as an early warning of upcoming epidemics using in data-poor scenarios but further investigation is necessary to inform surveillance efforts precisely.

This study was limited to synthetic data. The model was purposefully kept simple to ensure the genericity of the results. The emergence of the virus was driven by biting rates increasing in response to increasing temperatures over several years. However, temperature has an impact on other parameters, such as mosquito population dynamics parameters and the extrinsic incubation period. Previous research found that resilience indicators are robust to the choice of bifurcation parameter as long as the change is gradual [46]. Further, seasonal fluctuations were omitted, and the emergence of WNV was driven only by a gradual increase of $R_0$ over several years. Previous research with univariate indicators found a limited impact of seasonal fluctuations on the prediction performance of resilience indicators, even for fluctuations of high amplitude relative to the between-year increase [44]. Similarly, overdispersion was omitted in this model but previous research found that it has a limited impact on the performance of resilience indicators [45]. Although indicators of resilience loss hold potential to improve the current practice of early warning for vector-borne diseases, these findings should be confirmed in datasets generated with complex models, but also real data specific to a certain pathogen.

Compared to univariate indicators, multivariate indicators require sampling of many data sources simultaneously. This is complicated by cost and logistic limitations, but also coordination of the various authorities. The different data sources are often monitored by different institutions, such as public health authorities for human cases, veterinary authorities for domestic animals, and environmental authorities for wildlife. Coordinating the monitoring of different authorities and sharing data between the different institutions is sometimes challenging [47]. Additionally, costs and logistics limit the monitoring of wild birds and mosquitoes, and it is thus impossible to cover a whole, often heterogeneous area. This was omitted in the model, which assumed homogeneous mixing as well as homogeneous spatial coverage of wildlife data. In the case of spatial disparities in transmission, it is essential to consider high-risk zones when choosing where to execute the monitoring [48]. For instance, West Nile transmission can be affected by the local ecological context, such as the availability and abundance of different (competent and non-competent) hosts and vectors [32], leading to spatial heterogeneities in transmission potential [49]. Finally, some data sources studied here inherently contain biases that were not accounted for in the model. The model assumed that the number of infected mosquitoes was reported over time. In practice, the number

of infected mosquito pools is usually reported, reflecting the presence or absence of infected mosquitoes in a given area and not the number of infected mosquitoes [13]. Seroprevalence data is primarily informative about the past state of infection and could introduce biases if not sampled at regular intervals [26,50]. Additionally, when multiple similar pathogens are circulating simultaneously, antibodies targeting one pathogen may also react to a different, similar pathogen in tested individuals. This phenomenon, called cross-reactivity, can lead to a non-random overestimation of seroprevalence in areas where several pathogens are co-circulating. The dynamics of the co-circulating pathogen could be captured in the seroprevalence data, potentially introducing biases when measuring resilience [51]. Infected mosquito pools and seroprevalence data should thus be used with caution. While multivariate indicators have the potential to solve the challenge of large data requirements by monitoring several data sources instead of focusing on one sole source of information, many considerations remain in designing adapted surveillance programs. The final decision should be based on the specific challenges of a given area and pathogen, as well as whether the additional effort is worth the gain in prediction performance.

Finally, this study focused on the use of resilience indicators as an anticipation method, which is based solely on epidemiological time series and require prior detection of the virus. Although resilience indicators hold the advantage of being generic and model-free, and thus do not require model-fitting, other methodologies have been developed relying on environmental drivers such as temperature and precipitation as an early-warning indicator for West Nile virus outbreaks as well as for other mosquito-borne disease outbreaks [52–54]. A combination of such methodologies with resilience indicators could be envisioned to leverage the advantages of both methodologies.

## Conclusion

This study showed that multivariate indicators of resilience outperform univariate indicators in anticipating future outbreaks of mosquito-borne diseases, especially in data-poor contexts. As high granularity and quality of required data limit the use of resilience indicators in infectious disease epidemiology in practice, multivariate indicators hold the potential to improve the prediction performance of resilience indicators while dealing with data-poor contexts. However, adapting the surveillance programs to collect the relevant data for multivariate indicators of resilience entails new challenges related to costs, logistic ramifications and coordination of different institutions involved in surveillance. Ultimately, the aim is to inform effective monitoring strategies, and this study represents an initial step towards that objective.

## Supporting information

**S1 Text. Supplementary text, supplementary table A and supplementary figures A to AD.**
(PDF)

## Acknowledgments

We thank Marten Scheffer for the insightful discussions, his help with interpreting the results, and his contribution to the funding acquisition. We thank Els Weinans for providing her implementation of the multivariate indicators of resilience and taking the time to clarify some details about the methodology. We also thank the members of Work Package 3 of the One Health Pact consortium, Mariken de Wit, Afonso Dimas Martins, Mart de Jong, Hans Heesterbeek, and Martha Dellar, for their interest in the project and suggestions to improve this work.

## Author contributions

**Conceptualization:** Clara Delecroix, Quirine ten Bosch, Egbert H van Nes, Ingrid A van de Leemput.

**Data curation:** Clara Delecroix.

**Formal analysis:** Clara Delecroix.

**Funding acquisition:** Quirine ten Bosch, Ingrid A van de Leemput.

**Methodology:** Clara Delecroix, Quirine ten Bosch, Egbert H van Nes, Ingrid A van de Leemput.

**Supervision:** Quirine ten Bosch, Egbert H van Nes, Ingrid A van de Leemput.

**Validation:** Clara Delecroix, Quirine ten Bosch, Egbert H van Nes, Ingrid A van de Leemput.

**Visualization:** Clara Delecroix.

**Writing – original draft:** Clara Delecroix.

**Writing – review & editing:** Clara Delecroix, Quirine ten Bosch, Egbert H van Nes, Ingrid A van de Leemput.

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
