## [Decision Letter · Decision Letter 0]

17 Apr 2025

PCOMPBIOL-D-24-02124

Multivariate resilience indicators to anticipate vector-borne disease outbreaks: a West Nile virus case-study

PLOS Computational Biology

Dear Dr. Delecroix,

Thank you for submitting your manuscript to PLOS Computational Biology. After careful consideration, we feel that it has merit but does not fully meet PLOS Computational Biology's publication criteria as it currently stands. Therefore, we invite you to submit a revised version of the manuscript that addresses the points raised during the review process.

Please submit your revised manuscript within 60 days Jun 17 2025 11:59PM. If you will need more time than this to complete your revisions, please reply to this message or contact the journal office at ploscompbiol@plos.org. Please include the following items when submitting your revised manuscript:

We look forward to receiving your revised manuscript.

Kind regards,

Juliette Paireau

Academic Editor

PLOS Computational Biology

Roger Kouyos

Section Editor

PLOS Computational Biology

**Additional Editor Comments :**

The Authors are expected to address all the points raised by the Reviewers. In particular, the authors should give more details on the definition of resilience indicators (reviewers #1 and #3) and clarify the methodology used for performance assessment (reviewer #2). In addition, the code should be made available in the provided Github repository.

**Journal Requirements:**

1) Please upload all main figures as separate Figure files in .tif or .eps format. For more information about how to convert and format your figure files please see our guidelines: 

2) We have noticed that you have uploaded Supporting Information files, but you have not included a list of legends. Please add a full list of legends for your Supporting Information files after the references list.

3) Some material included in your submission may be copyrighted. According to PLOSu2019s copyright policy, authors who use figures or other material (e.g., graphics, clipart, maps) from another author or copyright holder must demonstrate or obtain permission to publish this material under the Creative Commons Attribution 4.0 International (CC BY 4.0) License used by PLOS journals. Please closely review the details of PLOSu2019s copyright requirements here: PLOS Licenses and Copyright. If you need to request permissions from a copyright holder, you may use PLOS's Copyright Content Permission form.

Potential Copyright Issues:

i) Figures 1A, and 3. Please confirm whether you drew the images / clip-art within the figure panels by hand. If you did not draw the images, please provide (a) a link to the source of the images or icons and their license / terms of use; or (b) written permission from the copyright holder to publish the images or icons under our CC BY 4.0 license. Alternatively, you may replace the images with open source alternatives. See these open source resources you may use to replace images / clip-art:

4) Please amend your detailed Financial Disclosure statement. This is published with the article. It must therefore be completed in full sentences and contain the exact wording you wish to be published.

2) State what role the funders took in the study. If the funders had no role in your study, please state: "The funders had no role in study design, data collection and analysis, decision to publish, or preparation of the manuscript.".

**Reviewers' comments:**

Reviewer's Responses to Questions

**Comments to the Authors:**

**Please note that one of the reviews is uploaded as an attachment.**

Reviewer #1: The manuscript provides an interesting modelling analysis aiming at evaluating the efficacy of resilience indicators to anticipate a vector borne outbreak, using synthetic West Nile virus epidemiological data. While finding the study well written and of interest, some lacking details and some doubts prevent me from fully appreciating it.

Major points

1. Throughout the manuscript the term “incidence” is used but I believe you actually mean prevalence. Incidence is defined as the number of new infections per unit of time, whereas prevalence is the fraction of infectious individuals, which is what is typically observed (for instance, analyzing mosquito pools provide an indication of the mosquito prevalence). Compartments MI, BVI, BHI, HI and EI denote the number of infectious individuals at time t, so again to me this is prevalence and not incidence. Please check carefully what you mean and amend the text accordingly.

2. Figure 1B. Why is there a sudden decrease in the number of infectious mosquitoes when R0 becomes greater than 1? This is rather counterintuitive. Plots show a single simulation, right? They are not the average of the 100 simulations, right?

3. I am not familiar with resilience indicators hence it is not clear to me what the authors mean and use. I would have expected some equations and more details illustrating what they are and how they are computed but these important points are currently missing. Please add some text, some examples and some equations (you can place them in the Appendix).

4. Another limitation which I believe needs to be acknowledged is that in your simulations R0 reaches the threshold value 1 in about 8 years, whereas in reality this occurs within a few weeks between spring and summer.

Minor points

1. Lines 49-51. Check this sentence.

2. Please describe WNV upon its first mention (line 79, you can move lines 114-116).

3. Lines 120-122. In the case of WNV, birds are also monitored by trapping and shooting them, see for instance [1].

4. Since you assume that all mosquitoes are non-diapausing (δM=1) you might consider removing this parameter to simplify the model.

5. Figure 1A. Since visible birds can either die or recover from the infection, I think there should be two arrows connecting BVI to BVR or BVD.

6. Please provide the unit (daily?) of the rates shown in Table S1.

7. Line 193. I assume you mean absence of infected mosquitoes and birds, please remark this (you might still have WNV circulating in mosquitoes but have no infectious birds for some time).

8. Lines 290-292. I understand that if the indicator signals a positive trend but R0<1 then this is a false positive, am I right?

9. Figure 2. Are these single stochastic simulations? Is recovery time expressed in days?

10. As WNV transmission is affected by several environmental drivers, there are many abiotic indicators (e.g. spring temperature) which might provide some kind of early warning indicator even before detecting the virus, which conversely is needed to use the indicators proposed in the manuscript. Please add some text to discuss this, with some references (e.g. [2–4]).

Bibliography

1. De Nardi A, Marini G, Dorigatti I, Rosà R, Tamba M, Gelmini L, et al. Quantifying West Nile virus circulation in the avian host population in Northern Italy. Infect Dis Model. 2025;10: 375–386. doi:10.1016/j.idm.2024.12.009

2. Marini G, Manica M, Delucchi L, Pugliese A, Rosà R. Spring temperature shapes West Nile virus transmission in Europe. Acta Trop. 2021;215: 105796. doi:10.1016/j.actatropica.2020.105796

3. Farooq Z, Rocklöv J, Wallin J, Abiri N, Sewe MO, Sjödin H, et al. Artificial intelligence to predict West Nile virus outbreaks with eco-climatic drivers. Lancet Reg Health - Eur. 2022;17: 100370. doi:10.1016/j.lanepe.2022.100370

4. Moirano G, Fletcher C, Semenza JC, Lowe R. Short-term effect of temperature and precipitation on the incidence of West Nile Neuroinvasive Disease in Europe: a multi-country case-crossover analysis. Lancet Reg Health - Eur. 2025;48: 101149. doi:10.1016/j.lanepe.2024.101149

Reviewer #2: The review is uploaded as an attachment.

Reviewer #3: This manuscript addresses a central problem of infectious disease surveillance—low and biased reporting—which may be particularly severe in multi-host vector-borne systems such as West Nile virus (WNV). The authors approach this issue using ideas from complex systems and time-series analysis related to signals of resilience. Specifically, the manuscript tests the use of multivariate resilience indicators using a model-generated synthetic dataset for WNV.

A key result, to me, is that multivariate indicators outperformed univariate indicators when sampling resolution and observation probability were reduced.

Major Comments

1. I suggest more technical precision and rigor in defining indicators of resilience. For example, around line 71, autocorrelation and variance are introduced as resilience indicators. What is the statistical test for slowing down when calculating variance and autocorrelation statistics in a time series over a moving window? Similarly, the sentence beginning on line 90 about multivariate indicators of resilience could use a mathematical definition or a “For instance…”

2. I’d like to see the differential equations for the model at the top of the Methods. “Adapted from (28)” isn’t helpful—even if the reader is familiar with reference 28—because we don’t know what “adapted” means. Just state the model.

3. For variance as an indicator, how does the approach deal with mean-variance scaling? How does this approach perform at different population sizes? What happens to the differential performance of multivariate versus univariate resilience indicators as population sizes become very small or very large? (By manipulating population size, I mean manipulating the overall magnitude of the time series.)

4. The monitoring scenarios assume a Poisson distribution. But infectious disease incidence data is often overdispersed and conforms better to a negative binomial distribution. I would be curious how the results change when using a negative binomial distribution, especially across different levels of overdispersion.

5. The perturbation-recovery experiments measure the rate of return to the DFE in an exponential model. But the introduction focuses on using variance and autocorrelation to measure resilience. Sorry if I missed it, but I suggest the authors add something to the introduction to discuss this return-time measure. The authors may be interested in this paper, which describes a related approach and could serve as a useful complement to the next-generation method for calculating R₀:

Neubert, M. G., & Caswell, H. (1997). Alternatives to resilience for measuring the responses of ecological systems to perturbations. Ecology, 78, 653–665.

6. Figure 2: I was expecting averages and credible intervals over replicate simulations. I suggest the authors include those and/or clarify what the figure shows and why, particularly with respect to statistical inference. Based on what's currently shown, it seems likely the replicates behave similarly, but this should be stated.

7. Figures 4 and 5: I'm not sure how to interpret changes in AUC as a function of variables such as sampling resolution and feeding preference. Why do AUC values fluctuate—for example, increasing slightly rather than decreasing—when sampling resolution goes from weekly to biweekly? When should such fluctuations be interpreted as meaningful trends, and when are they considered negligible? This part of the analysis may require a clearer statistical framework and potentially more replicate simulations.

Minor Comments

Line 69: “recovers slower” → “recovers more slowly”

There's a discontinuity between paragraphs 1 and 2 of the Introduction. The first is about resilience indicators and critical slowing down; the second jumps to multi-disease surveillance. Consider adding a transitional sentence to bridge these topics.

The paragraph starting on line 102 is important in making the case for multivariate resilience indicators for monitoring complex systems but needs some tightening. The sentence on line 104 could be removed or revised; complex systems by definition have multiple interacting components, and there is almost always a choice of where to direct monitoring. Similarly, the next sentence—"complex systems must have these types of interactions"—could be made more precise or omitted.

The Methods section switches to past tense after the first paragraph. I suggest using present tense throughout for consistency.

The term “generic bird” doesn’t have a clear biological meaning. Consider omitting “generic” or replacing with something like “representative bird species” or simply “bird.”

Line 177: The discrete events simulated by a Gillespie algorithm do not necessarily map to a branching process. Regarding why the Gillespie algorithm is used, I suggest wording like: “to model stochasticity arising from random variation in the timing of infection and removal events,” and omitting the reference to branching processes.

Line 261: For reproducibility, I suggest stating the optimal bandwidth used for detrending the emergence time series, rather than citing a prior paper without specifics.

Line 267: I suggest listing the indicators used at the top of the paragraph before discussing them in more detail. Many readers may not be familiar with the set used by Weinans et al. (2021).

The citation style shifts from numeric to author-date format in the Discussion. This should be made consistent.

Line 325: What does “no meaningful difference” mean statistically? Please clarify the basis for this claim.

Line 408: I suggest avoiding priority claims like “Our study was the first…” These are difficult to verify and don’t strengthen the results or their interpretation. Instead, focus on the specific contributions and implications for epidemic modeling and surveillance.

**Have the authors made all data and (if applicable) computational code underlying the findings in their manuscript fully available?**

Reviewer #1: **No: ** There is no available code in the provided github repository.

Reviewer #2: None

Reviewer #3: None

PLOS authors have the option to publish the peer review history of their article (what does this mean?). If published, this will include your full peer review and any attached files.

Reviewer #1: No

Reviewer #2: No

Reviewer #3: No

**Figure resubmission:**
---

## [Decision Letter · Decision Letter 1]

31 Aug 2025

PCOMPBIOL-D-24-02124R1

Multivariate resilience indicators to anticipate vector-borne disease outbreaks: a West Nile virus case-study

PLOS Computational Biology

Dear Dr. Delecroix,

Thank you for submitting your manuscript to PLOS Computational Biology. In light of the reviewers' assessments, we are likely to accept this manuscript for publication, provided that you revise it in accordance with the reviewers' recommendations.

Please submit your revised manuscript within 30 days Oct 31 2025 11:59PM. If you will need more time than this to complete your revisions, please reply to this message or contact the journal office at ploscompbiol@plos.org. Please include the following items when submitting your revised manuscript:

We look forward to receiving your revised manuscript.

Kind regards,

Juliette Paireau

Academic Editor

PLOS Computational Biology

Roger Kouyos

Section Editor

PLOS Computational Biology

**Journal Requirements:**

1) Please ensure that the affiliations of the authors listed on the manuscript title page (CITY) do exactly match with the affiliations provided in the online submission form

NOTE: Affiliations should include a department (if applicable), an institution, a CITY, and a country

**Reviewers' comments:**

Reviewer's Responses to Questions

**Comments to the Authors:**

Reviewer #1: Thank you for addressing all my previous comments and concerns

Reviewer #2: Thank you for your detailed, point by point response. All my previous concerns have been fully adressed.

I have only one minor comment: l364 the scenario with the 0.56 AUC seems to correspond to the Anthro-Equine scenario with explained variance. Please double check and correct if necessary.

Reviewer #3: Review of PCOMBIOL-D-25-012124_R1

The rebuttal letter is adequate, and this version of the manuscript is improved. I have one major suggestion which is to incorporate more of the authors’ responses to the reviewers into the manuscript. For instance, Reviewer #2 section 6 questions for authors. Readers may have these questions as well.

Minor comments:

Line 99 does this phrase need a comma? “mean, autocorrelation and variance”

Line 291 a window width of 50% the length of the time series seems rather large. Could the authors remark here on: 1) why they made it so wide and 2) how they handled missingness when the window extended beyond the beginning or end of the time series, which would occur often given then wide window?

Line 314 This is not correct for PCA. Suggest: “The maximum eigenvalue of the covariance matrix used in PCA is equal to the amount of variance explained by the first principal component.”

**Have the authors made all data and (if applicable) computational code underlying the findings in their manuscript fully available?**

Reviewer #1: None

Reviewer #2: Yes

Reviewer #3: Yes

PLOS authors have the option to publish the peer review history of their article (what does this mean?). If published, this will include your full peer review and any attached files.

Reviewer #1: No

Reviewer #2: No

Reviewer #3: No

**Figure resubmission:**
---

## [Editor Report · Decision Letter 2]

6 Oct 2025

Dear MSc Delecroix,

We are pleased to inform you that your manuscript 'Multivariate resilience indicators to anticipate vector-borne disease outbreaks: a West Nile virus case-study' has been provisionally accepted for publication in PLOS Computational Biology.

Best regards,

Juliette Paireau

Academic Editor

PLOS Computational Biology

Roger Kouyos

Section Editor

PLOS Computational Biology

---

## [Editor Report · Acceptance letter]

PCOMPBIOL-D-24-02124R2

Multivariate resilience indicators to anticipate vector-borne disease outbreaks: a West Nile virus case-study

Dear Dr Delecroix,

I am pleased to inform you that your manuscript has been formally accepted for publication in PLOS Computational Biology. Your manuscript is now with our production department and you will be notified of the publication date in due course.

With kind regards,

Zsofia Freund
